# Occurrence of Antibiotic-Resistant *Staphylococcus* spp. in Orange Orchards in Thailand

**DOI:** 10.3390/ijerph19010246

**Published:** 2021-12-27

**Authors:** Siwalee Rattanapunya, Aomhatai Deethae, Susan Woskie, Pornpimol Kongthip, Karl R. Matthews

**Affiliations:** 1Department of Public Health, Faculty of Science and Technology, Chiang Mai Rajabhat University, Chiang Mai 50300, Thailand; 2Department of Biology, Faculty of Science and Technology, Chiang Mai Rajabhat University, Chiang Mai 50300, Thailand; aomhatai_dee@cmru.ac.th; 3Department of Public Health, Zuckerberg College of Health Sciences, University of Massachusetts Lowell, Lowell, MA 01854, USA; susan_woskie@uml.edu; 4Department of Occupational Health and Safety, Faculty of Public Health, Mahidol University, Bangkok 10400, Thailand; pornpimol.kon@mahidol.ac.th; 5Department of Food Science, Rutgers, The State University of New Jersey, New Brunswick, NJ 08901, USA; bijan@sebs.rutgers.edu

**Keywords:** antimicrobial resistance, *Staphylococcus* spp., orange orchards

## Abstract

Background: The widespread indiscriminate application of antibiotics to food crops to control plant disease represents a potential human health risk. In this study, the presence of antibiotic-resistant staphylococci associated with workers and orange orchard environments was determined. A total of 20 orchards (orange and other fruits) were enrolled in the study. Trees in the orange orchards were treated with ampicillin on a pre-determined schedule. Environmental samples (n = 60) included soil, water, and oranges; 152 hand and nasal samples were collected from 76 healthy workers. Antibiotic susceptibility profiles were determined for all staphylococcal isolates. Results: This investigation revealed that of the total *Staphylococcus* spp. recovered from the orange orchard, 30% (3/10) were resistant to erythromycin, 20% (2/10) were resistant to ampicillin, and 20% (2/10) resistant to both erythromycin and ampicillin. Conclusion: The application of antibiotics to orange trees in open production environments to halt the spread of bacterial disease presents risks to the environment and creates health concerns for Thai farmers using those agents. ARB on crops such as oranges may enter the global food supply and adversely affect public health.

## 1. Introduction

Multidrug-resistant bacteria rank among the world’s most important public health problems of the 21st century. The World Health Organization (WHO) suggests that if the world neglects taking action, then we are heading to a post-antibiotic era, in which common infections and minor injuries will result in death [1]. Currently, antimicrobial resistance (AMR) trends support the perspective of the WHO that worldwide, at least 700,000 people die each year from common diseases, including respiratory tract infections, sexually transmitted infections, urinary tract infections, and foodborne illnesses [2]. Annually, more than two million people in the United States suffer from illnesses caused by antibiotic-resistant bacteria [ARB] [3]. A wealth of literature suggests that the ARB crisis is accelerated by the overuse and misuse of antibiotics in human medicine. Considerable attention has focused on the prevalence of ARB associated with food-producing animals and their environment, including commercial farms, feedlots, processing plants, and packing plants, since antibiotics are directly used for growth promotion and the prevention of diseases in food-producing animals [4,5,6,7,8]. The use of antibiotics and the spread of ARB associated with fruits (whether treated with antibiotics) through the global food supply is often overlooked [9]. ARB and antibiotic residues may also accumulate in the agriculture production environment, potentially adversely affecting farmworkers and consumer health.

According to regulatory agencies in the US and Britain, certain antibiotics (streptomycin, oxytetracycline) are permissible for use on production crops, including oranges, stone tree fruit, and pome fruit [10]. Gentamicin is used in the treatment of animal and human diseases but has been used to control plant diseases [11]. This is particularly true in Brazil, Korea, Thailand, and China for the control of citrus greening disease or Huanglongbing (HLB) and lettuce diseases [11,12]. In the United States, an alarm was raised for the spraying of antibiotics in open production environments to halt the spread of crop associated bacterial disease [13]. Indeed, from 1970 to the present, antibiotics such as ampicillin, amoxicillin, and tetracycline have been permitted for the prophylactic treatment of bacterial diseases in plants by *Candidatus* Liberibacter asiaticus or HLB in citrus via a graft-based chemotherapy method [14,15,16]. In Thailand, farmers or orchard operators regularly use capsule forms of ampicillin, amoxicillin, and tetracycline for the treatment of HLB. The recommended treatment dose of ampicillin is 12,500–25,000 ppm (i.e., achieved by adding 50 ampicillin 250 mg or 500 mg capsules per 1 L of water) through injection into tree trunks approximately three to four times per year [17,18]. Uncontrolled use of various antibiotics to treat HBL was encouraged by some orchard operators, academics, and government agencies. Orchard owners purchased the antibiotics directly from retail pharmacies or agrochemical suppliers [19]. Prevalence of ARB in animal production has been reported [19,20,21,22], an absence of literature is available on ARB associated with orange orchards.

Staphylococci are Gram-positive bacteria associated with the respiratory tract and skin of humans and animals. *Staphylococcus aureus* and members of the *S. intermedius* group are the clinically most important coagulase-positive staphylococci in human and veterinary medicine, respectively. Dozens of coagulase-negative staphylococcal species have been described as colonizers of the skin and mucous membranes and as food- associated saprophytes [23,24]. Most of them are less frequently involved in clinically manifested infections; however, in particular species of the S. epidermidis group account significantly for foreign body-related infections [24,25]. Multidrug-resistant (MDR) *Staphylococcus* spp. exhibited resistance rates to penicillin, ampicillin, and erythromycin of 96.6%–00%, 96.6%–8%, and 50%–7.1%, respectively [26,27,28]. A study in Thailand showed that the prevalence of community-acquired bacteraemia, healthcare-acquired bacteraemia, and hospital-acquired bacteraemia caused by MDR *S. aureus* was 8%, 28%, and 50%, respectively [29]. Antibiotic-resistant staphylococci have been isolated from the soil, surface water, wastewater, household surface dust, air, and a variety of crops intended for human consumption, as well as orange and apple juice products [30,31,32]. Antibiotic use in orange crop production may result in antibiotic residues in the environment and an increase in ARB entering the global food supply chain.

This study addresses the antibiotic susceptibility of *Staphylococcus* spp. Found in the environment (soil, water, and oranges surface) and workers (nasal and hand) from orange orchards.

## 2. Materials and Methods

### 2.1. Orange Orchards and Recruitment

Ten orange orchards and ten other fruit orchards (longan, and mango) located in northern Thailand were recruited from April 2020 to September 2020 using the selective inclusion criteria as follows: (1). Orange orchards were treated with antibiotics for more than 1 year, (2). Other fruit orchards received no antibiotics, (3). Orchards were not in proximity of potential sources of antibiotic contamination such as livestock production operations and hospitals, (4). Volunteer workers must be healthy, actively working in the orchards and have no history of illness requiring hospitalization in the inpatient department (IPD) or intensive care unit (ICU). Informed consent and assent were obtained from the orchard owners and workers.

### 2.2. Environmental Sample Collection

Samples of soil, water, and oranges were obtained. Samples were collected during periods when antibiotics were administered to orange trees. In brief, 10 soil samples from each orchard were collected from the antibiotic preparation area and under the orange tree canopy at pre-selected sites. Soil samples were collected by digging a V shape hole to a depth of approximately 15 cm. The hole was gouged on one side from top to bottom, and a 2–3 cm thick section of soil was collected and pooled. Ten water samples in a total volume of 1 L from different surface waters from each orchard were collected by immersing a sterile container in the water to a depth of 30 cm. The samples were then pooled [33]. Ten oranges/other fruits were collected into individual sterile bags containing diluent, placed into a cooler and transported back to the laboratory for microbiological analysis. Each bag was hand massaged for 60 s prior to sample collection [34].

### 2.3. Worker Samples Collection

Orange orchard workers (n = 44) and other fruit orchard workers (n = 32) were selected randomly to participate in the study. Hand and nasal samples were collected from each participant during the period when orchards were administering antibiotics to the orange trees. Hand samples were collected by placing a sterile grid 25 cm^2^ on the palm and back of the hand and swabbing the area with a sterile swab that had been immersed in a diluent. Swabs were then placed into sterile test tubes [35,36]. Nasal samples were collected using sterile cotton swabs at the same time as when hand samples were collected. A swab was inserted in the anterior nasal chamber (approximately 2 cm), rotated around the nasal mucosa, and placed back into a sterile test tube [37]. All samples were immediately placed in a portable cooler and transported back to the laboratory.

### 2.4. Bacterial Isolates

Samples were serial 1:10 diluted (in Phosphate-buffered saline), 0.2 mL of an appropriate dilution spread plated onto Mueller Hinton agar (MHA; Gibthai Co., Ltd., Chiang Mai, Thailand), and incubated at 37 °C for 24–48 h. Initial screening of the isolates was conducted based on colonial morphology on trypticase soy agar plates containing 5% sheep blood (TSA-SB; Biomedia [Thailand] Co. Ltd., Nonthaburi, Thailand), Gram stain, and catalase reaction. All tests were conducted in duplicate [37].

Isolates were identified using a biochemical tests kit (bioMerieux API, Marcy-l’Etoile, France). The total aerobic bacterial count was determined using the spread plate method [38].

### 2.5. Identification of Isolates

Species identification was based on 16S rDNA sequencing. The universal primers 27F (5′-AGAGTTTGATCCTGGCTCAG-3′) and 1492R (5′-GGTTACCTTGTTACGACTT-3′) were used for amplification [39]. The PCR conditions were as follows: one cycle of 95 °C for 5 min, followed by 35 cycles of denaturation at 94 °C for 30 s, primer annealing at 55 °C for 45 s, and extension at 72 °C for 2 min. A final extension step at 72°C for 10 min was then performed. Amplified products were isolated using 1% agarose gel electrophoresis, and purified products were submitted for DNA sequencing. DNA sequences were compared to a database of known 16S rRNA sequences using BLAST (blastn with default parameters) to identify the species of each isolate. (Macrogen, Seoul, Korea)

### 2.6. Antimicrobial Susceptibility Testing

Antimicrobial susceptibility tests were performed using the Kirby–Bauer disc diffusion method [38]. Susceptibility to ampicillin (10 μg), tetracycline (30 μg), and erythromycin (15 μg) was determined (Himedia, India). Only zones of complete inhibition by visual inspection were measured, recorded, and interpreted as susceptible (S), intermediate (I), or resistant (R).

### 2.7. Data Analysis

The antibiotic susceptibility of *Staphylococcus* spp. in each type of environmental sample were described by the number and percentage of positive samples.

## 3. Results

### 3.1. Demographics

Orange orchards were placed into one of three size ranges; <1.6 hectares (53.3%), >1.6 hectares to 3.2 hectares (30%), and >3.2 hectares (16.7%). Orange trees in all orchards were treated with ampicillin (12,500–15,000 ppm of ampicillin in water) for at least one year. Ampicillin was delivered through injection or injection drip into tree trunks approximately three to four times per year (Figure 1). In control orange orchards, no antibiotics were used. Orchard workers included owners and employees. Employees were temporary workers averaging 10.1 ± 8.1 years’ experience, working 7.1 ± 2.1 h/day, and 5.1 ± 1.9 days/week. Owners averaged 16.2 ± 11.9 years’ experience, working 7.1 ± 2.1 h/day, and 5.4 ± 2.2 days/week. Orange orchards included in the study were not in close proximity to livestock farming operations or hospitals that may serve as potential sources of antibiotic contamination.

### 3.2. Identification of Staphylococcus Species Isolated

A total of 212 samples were collected, which included 60 soil, water, and orange samples from orange orchards and non-orange fruit orchards; 152 hand and nose swab samples were collected from 76 healthy workers. A total of 42% (21/50) of bacterial isolates came from the environment, and 76% (31/41) of bacterial isolates were from workers. *Staphylococcus* spp. were recovered from 19% (6/31) of the environmental samples and 13% (4/21) of the worker samples. However, it is worth noting that *Staphylococcus* spp. were more prevalent in orange orchards than other fruit orchards (9:1). The staphylococci belonged to four species: *Staphylococcus epidermidis* (40%), *Staphylococcus arlettae* (30%), *Staphylococcus haemolyticus* (20%), and *Staphylococcus saprophyticus* (10%) (Table 1).

### 3.3. Antimicrobial Susceptibility

Antibiotic susceptibility was determined for each staphylococcal isolate (Table 2 and Appendix A). Most isolates (70%; 7/10) were resistant to at least one antibiotic. All staphylococcal isolates recovered from orange orchards exhibited resistance: 30% (3/10) resistant to erythromycin, 20% (2/10) resistant to ampicillin, and 20% (2/10) resistant to erythromycin and ampicillin (Figure 2). None of the isolates exhibited resistance to tetracycline. *Staphylococcus arlettae* isolated from water samples exhibited resistance to erythromycin and ampicillin. *Staphylococcus haemolyticus* and *Staphylococcus saprophyticus* recovered from environmental samples were resistant to erythromycin. *Staphylococcus epidermidis* isolated from workers exhibited resistance to ampicillin. In general, erythromycin resistance was associated with environmental isolates, while ampicillin resistance was associated with isolates recovered from the environment and workers.

## 4. Discussion

There is a significant knowledge gap concerning the use of antibiotics on edible fruit crops and the occurrence of ARB on those fruits. In the present study, *Staphylococcus* spp. resistant to erythromycin and ampicillin were only recovered from samples collected from orange orchards applying ampicillin to trees for the control of HLB. *Staphylococcus* spp. are associated with diseases of humans and animals and have been recovered from the environment (e.g., water and soil); therefore, isolation from an orange production environment is not unexpected. The recovery of antibiotic-resistant staphylococci from oranges and orange orchard environments raises concern for the spread of ARB through the food supply chain. The occurrence of antimicrobial-resistant microorganisms associated with fresh fruits or their production environment is of human health importance. For example, *Aspergillus fumigatus* resistance to all triazole antifungals recovered from patients likely originated in the environment [40].

The unregulated application of antibiotics may result in increased populations of ARB and antibiotic residues on crops intended for human consumption resulting in the spread of antimicrobial resistance (AMR) and exacerbating a global health crisis. Thai mandarin orange growers indicated they adjusted HLB antibiotic treatment, antibiotic concentration, volume, frequency, route of administration, and combination of antibiotics based on outcomes rather than following science-based recommendations [18]. Orange growers participating in the present study used ampicillin for the treatment of HLB, which is not approved for use, for example, in the United States and Britain. Very few studies have addressed the issues of antibiotic use on food crops and ARB on antibiotic-treated crops or the surrounding environment. Parameters influencing the mobility and stability of antimicrobials in the environment must be addressed to protect the global food supply and human health.

The *Staphylococcus* spp. isolated in the present study are classified as coagulase-negative staphylococci (CoNS) and are associated with nosocomial infections [24]. The results of this study are consistent with previous reports in which *S. epidermidis*, *S. arlettae*, *S. haemolyticus*, and *S. saprophyticus* recovered from non-healthcare settings [41,42] and healthcare settings [43,44] were often resistant to multiple antibiotics (e.g., penicillin, erythromycin, amoxicillin, and ampicillin). Unlike meat and poultry, which are thermally processed prior to cooking, oranges are consumed raw, and therefore ARB is not inactivated. ARB on the orange surface can cross-contaminate the edible portion of the orange during peeling. The present study did not investigate the potential dissemination of antibiotic residues associated with oranges exposed to antibiotics. Although coagulase-negative staphylococci are not considered foodborne pathogens, they may play an important role in the spread of antibiotic resistance genes in community and hospital environments.

## 5. Conclusions

The ramifications of the direct application of antibiotics to food crops such as oranges and the emergence and spread ARB on human health are not fully appreciated. The application of antibiotics to orange trees in open production environments to halt the spread of the bacterial disease of a crop presents risks to the environment and creates health concerns for farmers using those agents. The present study focused on Gram- positive bacteria exhibiting multi-antibiotic resistance. Gram-negative bacteria are commonly isolated from fresh fruits and vegetables and therefore should be included in future investigations of ARB. ARB and antibiotic residues on crops such as oranges may enter the global food supply and adversely impact human health. The presence of ARB on foods that will not undergo a process to inactivate the bacteria prior to consumption may represent a greater risk for the spread of ARB and antibiotic resistance genes.

## Figures and Tables

**Figure 1 ijerph-19-00246-f001:**
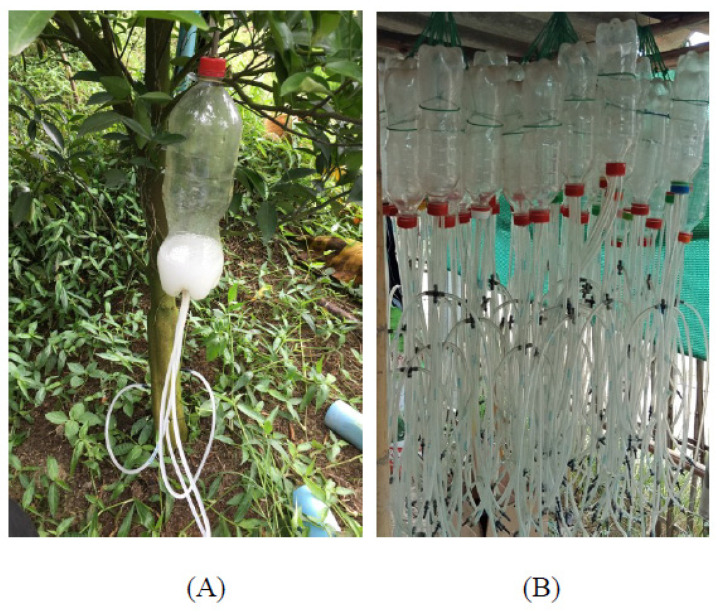
Ampicillin being administered to an orange tree (**A**). Drip bottles used to administer ampicillin (**B**).

**Figure 2 ijerph-19-00246-f002:**
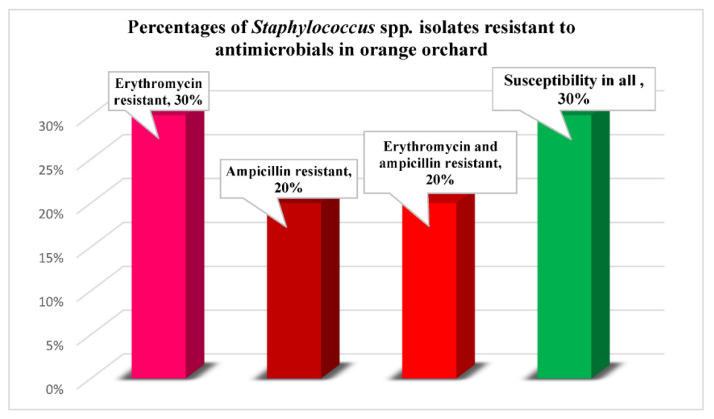
Percentages of *Staphylococcus* species recovered from orange orchards resistant to antibiotics.

**Table 1 ijerph-19-00246-t001:** Identification of *Staphylococcus* species isolated from environmental and worker samples from orange orchards or other fruit orchards.

*Staphylococcus* Species	Orange Orchard		Other Fruit Orchard		Total	
	Env.	Worker	Env.	Worker	Env.	Worker
*Staphylococcus epidermidis*	-	3 ^h,h,n^	-	1 ^n^	-	4
*Staphylococcus arlettae*	2 ^w,w^	1 ^n^	-	-	2	1
*Staphylococcus haemolyticus*	1 ^os^	1 ^n^	-	-	1	1
*Staphylococcus saprophyticus*	1 ^s^	-	-	-	1	-

Env. = environmental; ^h^ = hand; ^n^ = nose; ^os^ = orange surface; ^s^ = soil; ^w^ = surface water.

**Table 2 ijerph-19-00246-t002:** Antibiotic susceptibility of staphylococci isolated from the environment and workers.

	Susceptibility *		
Staphylococcal Isolate	Ampicillin	Erythromycin	Tetracycline
*Staphylococcus epidermidis* 01 008h1g2	R	S	S
*Staphylococcus epidermidis* 008h1g3	R	S	S
*Staphylococcus epidermidis* 009n1g1	S	S	S
*Staphylococcus epidermidis* 016h1g2	S	S	S
*Staphylococcus arlettae* 001wg6	R	R	S
*Staphylococcus arlettae* 003wg1	R	R	S
*Staphylococcus arlettae* 006n2g1	S	R	S
*Staphylococcus haemolyticus* 001n2b1	S	S	S
*Staphylococcus haemolyticus* 002worg2	S	R	S
*Staphylococcus saprophyticus* 004soig1	S	R	S

* susceptible (S), intermediate (I) or resistant (R).

## Data Availability

Not applicable.

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
