# Peer review of "Occurrence of Antibiotic-Resistant Staphylococcus spp. in Orange Orchards in Thailand"

_ijerph, 2021, doi:10.3390/ijerph19010246_

Round 1
Reviewer 1 Report
I have carefully reviewed the manuscript entitled Occurrence of Antibiotic Resistant Staphylococcus spp. in Orange Orchards in Thailand and I have the following recommendations/
Abstract:
Line 16: Please remove the word background.
Line 21: Please keep the font size according to the journal requirement.
Introduction:
Line 31: The World Health Organization (WHO) suggests that with-out urgent action in a post-antibiotic era common infections and minor injuries will once again kill [1]. Please improve the sentence for better understanding.
Line 52: Please add some words regarding to remediation techniques and highlight the research gap for ubiquitous application of antibiotics.
Line 75: Please add objectives of the study.
Line 90: Please mention the sampling collection method in supporting information.
Line 96: 1 liter or litter? Please proofread this manuscript from native or English expert, remove all the sentence and grammar mistakes.
Line 142-152: Please keep the font size same and remove the extra space after full stop.
Conclusion: Please improve the conclusion.
References: Please use the same format for references.
Grammar: Major issue with the grammar, please revise.
Author Response
Dear, Reviewer
Please see the attachment.
Sincerely yours,
Siwalee R.

Reviewer 2 Report
The authors report on the presence of antibiotic-resistant staphylococcal species associated with workers and orange orchard environments. This study is of major interest since it helps to close gaps of our knowledge of antibiotic resistance pressure outside human and veterinary medicine. However, the manuscript needs some improvements.
Major comments
- L65-67: Please differ between S. aureus and coagulase-negative staphylococci.
- Materials and Methods: Which antibiotics have been given by the farmers, please specify.
- Results, L159-160: This sentence is completely misleading, please re-phrase. These are not the percentages of all bacterial isolates recovered.
- Results, L171ff: Concerning resistance, please differ between environmental samples and samples derived from the workers.
- Please correlate the resistances also with the kind of the farm investigated (using or not using antibiotics).
- Results: Although it is understandable that the authors restricted their study to the three chosen antibiotic compounds, a broader view on the resistance pattern would be of interest. I would recommend to include some more agents of human and veterinary interest (e.g. to detect possible co-selection pressure effects or general pollution of the environment by antibiotic agents)
- Results: It would be of interest whether the ampicillin resistance is due to the activity of beta-lactamases or by additional PBP2a (i.e., mecA/mecC-caused). Please check e.g. by respective PCR. This would be of interest, again e.g. to detect possible co-selection pressure effects.
- Discussion: Are there any data available on the concentration of administered antibiotics in harvested oranges or other fruits included? Please discuss.
Minor comments
- Throughout the text: The authors have to correct a lot of orthographic, grammar and stylistic mistakes. Please use the help of a native English speaker.
- Throughout the text: please do not italicize “spp.”
- Throughout the text: If a “Candidatus” designation is used, the term “Candidatus” has to be given in italics, but but not the genus name and/or the vernacular epithet
- excel table: Please give a headline explaining the content of this table
- excel table: Sample type No. 10, please translate into English; a bracket is missed (E15)
- excel table: Explain the abbreviations used in this table
- Please correct ref. no. 12 (Qiao et al.)
Author Response

(The authors gave the same response as above.)

Round 2
Reviewer 2 Report
L168 ff: I am not yet satisfied with the sentences about staphylococci in the Introduction section. Besides S. aureus and S. intermedius, there are also other coagulase-positive staphylococcal species and the clinical significance is still unclear. Thus, I would suggest to re-phrase it, e.g.: “Staphylococcus aureus and members of the S. intermedius group are the clinically most important coagulase-positive staphylococci in human and veterinary medicine, respectively. Dozens of coagulase-negative staphylococcal species have been described as colonizers of the skin and mucous membranes and as food-associated saprophytes (cite also here e.g. #40 Becker et al.). Most of them are less frequently involved in clinically manifested infections, however, in particular species of the S. epidermidis group account significantly for foreign body-related infections (cite e.g. again #40 and/or Heilmann et al., Are coagulase-negative staphylococci virulent? Clin Microbiol Infect. 2019;25:1071-1080.).”
Author Response
Dear, Reviewer
Thank you so much for your suggestion. Manuscript has been revised according to your suggestion.
“Staphylococcus aureus and members of the S. intermedius group are the clinically most important coagulase-positive staphylococci in human and veterinary medicine, respectively. Dozens of coagulase-negative staphylococcal species have been described as colonizers of the skin and mucous membranes and as food-associated saprophytes (23-24). Most of them are less frequently involved in clinically manifested infections, however, in particular species of the S. epidermidis group account significantly for foreign body-related infections (24-25).”
Sincerely yours,
Siwalee R.